# Research Progress in Enzymatically Cross-Linked Hydrogels as Injectable Systems for Bioprinting and Tissue Engineering

**DOI:** 10.3390/gels9030230

**Published:** 2023-03-15

**Authors:** Raquel Naranjo-Alcazar, Sophie Bendix, Thomas Groth, Gloria Gallego Ferrer

**Affiliations:** 1Centre for Biomaterials and Tissue Engineering (CBIT), Universitat Politècnica de València, 46022 Valencia, Spain; 2Department of Biomedical Materials, Institute of Pharmacy, Martin Luther University Halle-Wittenberg, Heinrich-Damerow-Strasse 4, 06120 Halle (Saale), Germany; 3Interdisciplinary Center of Material Research, Martin Luther University Halle-Wittenberg, 06120 Halle (Saale), Germany; 4Biomedical Research Networking Center on Bioengineering, Biomaterials and Nanomedicine, Carlos III Health Institute (CIBER-BBN, ISCIII), 46022 Valencia, Spain

**Keywords:** enzymatic hydrogels, bioprinting, tissue engineering, cross-linking reactions, injectable hydrogels

## Abstract

Hydrogels have been developed for different biomedical applications such as in vitro culture platforms, drug delivery, bioprinting and tissue engineering. Enzymatic cross-linking has many advantages for its ability to form gels in situ while being injected into tissue, which facilitates minimally invasive surgery and adaptation to the shape of the defect. It is a highly biocompatible form of cross-linking, which permits the harmless encapsulation of cytokines and cells in contrast to chemically or photochemically induced cross-linking processes. The enzymatic cross-linking of synthetic and biogenic polymers also opens up their application as bioinks for engineering tissue and tumor models. This review first provides a general overview of the different cross-linking mechanisms, followed by a detailed survey of the enzymatic cross-linking mechanism applied to both natural and synthetic hydrogels. A detailed analysis of their specifications for bioprinting and tissue engineering applications is also included.

## 1. Introduction

Hydrogels have been studied by many authors as 3D in vitro systems because, in addition to having all the characteristics of scaffolds (biocompatibility, biodegradability, stability, porosity and cytocompatibility), they offer better control of the spatial position of cells and facilitate cell–cell interaction [1]. Hydrogels are 3D networks of hydrophilic polymeric chains capable of absorbing significant amounts of water [2,3,4,5]. The highly hydrated environment (i.e., >90 wt. % water content) reproduces the aqueous environment of in vivo tissue systems and permits additional complexity via the encapsulation of cells and bioactive molecules [6]. For instance, in bioactive molecules, the incorporating of peptide domains, such as arginine–glycine–aspartic acid sequence (RGD), into the 3D matrix increases cellular adhesion [7] and mimics the situation of the native tissue.

A macromolecular network of hydrogels is formed by physical or chemical bonds of the hydrophilic polymeric chains at cross-linking points. The most important parameters used to characterize a network configuration are the polymer volume fraction in the swollen state (the amount of fluid retained by the hydrogel), the average molecular weight between cross-links (connected with the cross-linking density), and the distance between cross-links (mesh size) [7,8]. All these are related to each other, such that the cross-linking density can be used to control mechanical stiffness, swelling and mesh size, thus affecting cell encapsulation. The network mesh size controls the mass transport (oxygen, nutrients, biomolecules, etc.) through the hydrogel. When hydrogels are cross-linked in the presence of water, different porosities can be achieved and their swelling capacity and diffusivity are improved, to the detriment of stiffness. A suitable balance of all the properties must be reached for the tissue to be mimicked.

Apart from achieving the proper equilibrium of their properties, current studies focus on hydrogels that can vary their shapes when exposed to external stimuli, such as temperature, environmental pH, the concentration of dissolved ions, electric fields or shear stress. These are known as shape memory hydrogels (SMHs) or smart hydrogels [9] and are the center attention due to their being able to adopt many different shapes according to the required external factors.

Because of all these specific characteristics, hydrogels are widely considered an outstanding candidate for different tissue engineering applications [10,11,12]. Recent efforts have focused on the search for mild cross-linking methods that can take place in the presence of cells by means of non-cytotoxic reactions. These systems are called in situ gelling or injectable hydrogels, because the precursor solution containing the cells can be injected into the body in liquid form and becomes solid in the implantation site without damaging the host tissue or cells. They can be advantageous, since the cells can be homogeneously incorporated into the matrix and the scaffold is implanted by minimally invasive surgery, completely filling defects of any shape. Biodegradation kinetics is a crucial factor in this type of hydrogel since new tissue must be generated as the matrix degrades.

The additive manufacturing industry is incorporating cell-based polymer solutions into bioprinting processes to build personalized and complex scaffolds [13,14]. Bioprinters are composed of various nozzles by which a raw material is printed with a specific shape and solidifies outside the bioprinter to imitate an organ. The raw material used is called bioink, which is a combination of a hydrogel precursor solution (ink) and cells. Cross-linking outside the bioprinter can be done via different processes, such as the use of external light, or the combination of the bioink extruded from one nozzle with an ink printed by the following nozzle containing the cross-linker or a component that catalyzes the cross-linking. Different cross-linking mechanisms, bioinks and types of bioprinters have been designed to mimic customized patient defects or the hierarchical layered structures of tissues through a printed hydrogel.

In the present review, we describe the hydrogel enzymatic cross-linking mechanisms as they are the leading constructs for tissue engineering applications. We discuss different types of their physical and chemical in situ gelling/cross-linking, outlining their advantages and limitations, and conclude that enzymatic cross-linking is the optimal method. Different enzymes and their cross-linking mechanisms are explained in detail, as well as the natural and synthetic polymers used. Finally, we explain the distinct applications of these systems (enzymatic-mediated hydrogels) as injectable hydrogels and bioinks for 3D bioprinting.

## 2. In Situ Gelling Reactions

The cross-linking mechanism, leading to hydrogel formation, can be physical, chemical, or a combination of both. This classification is general for hydrogels, in the sense that it applies to all cross-linking methods including those that are cytotoxic and cannot take place in the presence of cells. We here describe both mechanisms and summarize the main in situ cross-linking methods (Table 1).

### 2.1. Physical Cross-Linking

The 3D network of physical hydrogels is created by reversible interactions, such as hydrogen bonds, ionic and Van der Waals forces. Covalent interactions do not take place. As chemical cross-linking agents are absent, physical hydrogels are easily compatible with biomolecules and living cells. However, as cross-linking points are formed by weak interactions, these hydrogels have poor stability and mechanical properties. Their reversibility can be advantageous in tissue engineering because they can react to local changes of pH or the concentration of substances in the body by dissolving and releasing encapsulated molecules. In applications in which stability can be a problem, chemical agents can be added to provide systems with both physical and chemical cross-links, forming what we can call dual hydrogel networks. In these cases, physical cross-linking is considered a form of pre-cross-linking previous to the chemical one.

Of the physical interactions, we here describe self-assembly, ionic interactions, thermal condensation, and stereo-complexation, with examples of in situ gelling hydrogels.

#### 2.1.1. Self-Assembly

Molecular self-assembly is based on non-covalent interactions and weak forces such as hydrogen bonds, hydrophobic and hydrophilic interactions, Van der Waals forces and π-stacking (Figure 1a). Due to their shear thinning and self-healing properties, i.e., they exhibit low viscosity under shear stress and once removed they gel, these systems have been considered for bioinks and injectable hydrogels. The challenge is the self-healing time, because if it is too long the encapsulated cells can be affected. Some authors have focused on these systems [15,16,17] after Samuel Stupp showed their potential [18,19].

#### 2.1.2. Stereo-Complexation

Stereo-complexation occurs when oligomers of opposite chirality, such as d- and l-lactic acid, are in contact. The cross-linking mechanism is based on the interaction of two different polymer chains, one modified with poly(l-lactic acid) (PLLA) and the other with poly(d-lactic acid) (PDLA). The reaction of PLLA and PDLA generates the cross-linking points and the resultant 3D matrix (Figure 1b). Many authors have considered PEG as the most suitable polymer for stereo-complexation hydrogels [20,21,22,23,24].

Despite the drawbacks of physical cross-linking, stereo-complexation allows the creation of complex structures that can be used for specific applications. For instance, cyclodextrins have been used for studying the solubilization of hydrophobic drugs thanks to their hydrophobic internal cavity [25].

#### 2.1.3. Ionic Interactions

Hydrogels are synthesized by bonding opposite charges in two possible ways: chelation (Figure 1c) or electrostatic interaction (Figure 1d). This can occur when polymer chains (backbone or lateral groups) have a net charge that attracts charged species of the opposite sign, forming insoluble complexes known as ionotropic hydrogels. These electrically charged polymers are called polyelectrolytes. Many natural polymers are negatively charged in aqueous solutions at physiological pH [7], such as hyaluronic acid [26,27], chondroitin sulfate [28], carrageenan [29], dextran [30], and alginate [31,32] due to the presence of carboxylate (COO^−^) or sulphate groups (SO_4_^2−^). On the other hand, positively charged polymers also exist, usually in acidic conditions, as is the case of chitosan and gelatin, in which the amino groups or their backbone protonate (NH^3+^). In this case, once the solution is prepared under acid conditions, various steps should be taken to achieve physiological pH and allow cell encapsulation.

#### 2.1.4. Thermal Cross-Linking

Some polymers, such as agarose, carrageenan, gelatin, elastin and collagen, change from a liquid to a solid state by varying their temperature. This type of hydrogel is called a thermosensitive hydrogel, in which gelation occurs by increasing or reducing the temperature. The transition temperature can be the upper critical solution temperature (UCST) or the lower critical solution temperature (LCST). For polymers with a UCST, gelation takes place at a temperature below the UCST [2], while for those with an LCST, gelation occurs at a temperature above the LCST (Figure 1e,f). Different polymers have been used as injectable hydrogels and bioinks based on thermal cross-linking, such as collagen [33], agar [34,35], agarose [36,37], gelatin [38,39], chitosan [40], and hyaluronan [41].

The main advantage of these systems is their reversibility in specific applications such as drug delivery. Once the hydrogel is inside the body, the matrix can be broken due to the body temperature releasing the drug.

### 2.2. Chemical Cross-Linking

Chemical cross-linking is based on covalent interactions, resulting in an irreversibly cross-linked network. This type of cross-linking provides stability to the hydrogel, which keeps its structure after in situ injection or printing. Many chemical cross-linking mechanisms are compatible with cell survival. We here summarize the main ones applied to hydrogels, including photo-induced, click chemistry and enzymatic cross-linking.

#### 2.2.1. Photo-Induced

In light-based chemistries, two polymer chains are covalently bonded after exposure to light irradiation in the presence of photo-initiators. Three reactions can be distinguished [6]: free-radical chain cross-linking, thiol-ene photo-cross-linking and photomediated redox cross-linking (Figure 2). Photo-induced cross-linking reactions offer fast cross-linking, spatiotemporal control over the reaction, room temperature conditions and maintenance of the hydrogels’ shape. However, the intensity of light, exposure time and photo-initiator concentration are crucial parameters that can damage cells. Long exposure times and high light intensities damage the cells’ DNA. UV light (290–320 nm) is suitable for inks that do not contain cells, but not for bioinks, so that visible light such as blue (405 nm) and green lights (505 nm) have been explored as alternatives, since their intensity is quite similar to that of UV, and they do not damage the cells [42]. The photo-initiator needs to be non-cytotoxic and to interact correctly with the light source. Several photo-initiators are commercially available, eosin being one of the most prominent candidates for use with visible light [3].The most common polymers used are PEG [43,44,45,46], gelatin [47,48,49], and hyaluronan [50,51].

#### 2.2.2. Click Chemistry

Click chemical cross-linking is performed by the reaction of two different mutually reactive groups, such as aldehyde and hydrazide [52]. Depending on the reactive couple and its concentration, a wide range of well-defined microenvironments for cells can be achieved for both bioinks and injectable hydrogels. Additionally, the reactions happen fast, spontaneously, and under mild conditions. In tissue engineering, the alkyne–azide cycloaddition reaction is not used, since copper, which is cytotoxic to cells, participates in the reaction. Copper-free click reactions and pseudo-click reactions are thus considered for hydrogel synthesis. Diels-Alder [53,54,55,56], SP-AAC [57,58,59,60], thiol-ene [61,62,63,64,65] and oxime [66,67,68,69] reactions are copper-free click mechanisms, while thiol-Michael addition [70,71,72,73,74] and aldehyde–hydrazide [75,76,77,78] are pseudo-click reactions (Figure 3).

#### 2.2.3. Enzymatic

Enzymatic-based hydrogels are covalently bonded by reactions that are catalyzed by specific enzymes according to the polymer used. For instance, for horseradish peroxidase (HRP)/H_2_O_2_ cross-linking, phenolic groups must exist in the polymer structure to create the 3D network. Enzymatic cross-linking is the most cytocompatible process, since no exogenous reagents are used [3]. All the enzymes used for injectable hydrogels and bioinks are commonly found in humans. The enzymatic cross-linking process can also take place under mild conditions such as neutral pH, physiological temperature, and an aqueous medium [79,80], which are compatible for cells. The catalyst remains mostly unaffected from the reaction. Since enzymes are proteins made of amino acids, it is important to consider the environmental conditions (e.g., pH, ionic strength, temperature, etc.) to use the enzyme, because protein unfolding will reduce or destroy the enzymatic activity [81,82]. Enzyme specificity is an advantage, as toxicity can be avoided, and certain enzymes react with certain polymers without generating cytotoxicity. Similarly, as in all chemical cross-linking, enzymatic cross-linking is fast-gelling and leads to hydrogels with good mechanical properties that can be controlled by the enzyme activity and concentration [83,84,85,86,87,88]. Despite being expensive and difficult to produce [52], enzymatic hydrogels provide a suitable microenvironment for cells, and are the most prominent mechanism for developing injectable hydrogels and bioinks. Their fast gelation is an advantage over other cross-linking reactions because it prevents cells sedimentation during the cross-linking and ensures that the bioink stays in shape.

The present study is focused on this type of chemical cross-linking, giving the most commonly used polymers in Section 3 and describing the chemical reactions that take place during cross-linking in Section 4. Section 5 and Section 6 describe two different applications in tissue engineering in the form of injectable systems and bioprinting, respectively.

**Table 1 gels-09-00230-t001:** Advantages and disadvantages of physical and chemical cross-linking.

Cross-Linking		Advantages	Disadvantages
Physical	Self-assembly	Reversible mechanism [A] [52] Compatibility with biological systems [B] [3] Shear-thinning [2] Self-healing [2]	Additional post-cross-linking [C] [42] Poor mechanical properties [D] [52] Prolonged self-healing [2]
	Ionic interactions	[A,B] [3,52] Working under mild conditions [89]	[C,D] [42,52] Exhaustive of the number of ions [42]
	Thermal cross-linking	[A,B] [3,52] Rapid reassembly to hydrogel [2] Work under physiological conditions [89]	[C] [42] Precise temperature for cell viability [2]
	Stereo-complexation	[A,B] [3,52]	[C,D] [42,52]
Chemical	Photo-induced	Stabilization of weak cross-linked hydrogels [3,6] Fast gelation [6] Spatiotemporal control of the reaction [42] Room temperature conditions [42]	Light irradiation may affect cells [3] Precise determination of photo-initiator, intensity light and exposure time [3,42]
	Click chemistry Diels-Alder SP-AAC Thiol-ene Oxime Thiol-Michael Aldehyde-hydrazide	Fast gelation (all mechanisms) [90] Mild conditions (all mechanisms) [2,90] Spontaneous reaction (all mechanisms) [90,91] Good mechanical properties (all mechanisms) [90] Not sensitive to oxygen or water (Thiol-ene) [90]	Long gelation without initiator (Diels-Alder) [92] Numerous steps for the cyclooctyne’s synthesize (SP-AAC) [90] Use of an initiator (Thiol-ene) [90] Basic pH could damage cells (Oxime) [93]
	Enzymatic	No exogenous reagents [3] Spontaneous reaction [79] Control over the reaction [79] Specificity [52] Fast gelation [89] Mild conditions [42,52]	Needs additional catalyst (enzyme): the activity can change during the storage of the stock solution [3] The costs of the enzyme are additional costs [52]

## 3. Biological and Synthetic Macromolecules for Enzyme-Cross-Linked Hydrogels

Some natural polymers obtained from extracellular matrix (ECM) sources are hydrophilic and can form hydrogels with high biocompatibility, biodegradability, and bioactivity. However, in many cases they require chemical modification to make them enzymatically cross-linkable. In this section, we summarize the main characteristics of the most popular naturally derived polymers that can give rise to enzymatic hydrogels in the two main groups: polysaccharides (alginate, chitosan, hyaluronic acid, chondroitin sulphate, dextran), whose monomeric units are described in Figure 4, and proteins (collagen, elastin, gelatin and silk). In Section 5, we describe the enzymatic cross-linking mechanism used for these macromolecules, with different examples of enzymatic cross-linking hydrogels.

### 3.1. Polysaccharides

Alginate is a linear polysaccharide copolymer composed of two monomers, (1-4)-linked β-d-mannuronic acid (M) and α-l-guluronic acid (G). These monomers are covalently linked resulting in GG, MM, and MG blocks. The blocks’ sequence when generating alginate’s structure depends on the source, which can be brown seaweed or bacteria such as Laminaria hyperborea and Laminaria japonica [94]. The distribution of the blocks in terms of M/G ratio has a strong influence on the chain configuration and hydrogel mesh size. Since alginate does not promote cell adhesion, it is usually modified with collagen, laminin or RGD sequences for tissue engineering [7].

Chitosan is a polycationic polysaccharide produced from the deacetylation of chitin [95], which is the main component of the exoskeleton of crustaceous water animals. Chitosan’s structure (C_6_H_11_O_4_N)m is composed of glucosamine repeating units, where the amino groups interact with specific components resulting in the desired scaffold. Due to the amino functional groups, chitosan is soluble at acid pH, being a polycationic polymer with a pH value of around 6. It is widely used in tissue engineering, especially in cartilage regeneration, as it is nontoxic, biodegradable, inhibits tumor cells, stimulates the immune system, accelerates wound healing, etc. [96].

Chondroitin sulfate (CS) is the most abundant glycosaminoglycan found in vertebrate and invertebrate extracellular matrixes [97]. Its linear polysaccharide structure is composed of repeating β-1,3-linked N-acetyl galactosamine (GalNAc) and β-1,4-linked d-glucuronic acid (GlcA) disaccharide. The carboxylic acid groups (COOH) present in the monomeric units are essential in the chemical modification of the CS, allowing subsequent gelation through different mechanisms. Unlike hyaluronic acid, chondroitin sulfate is a sulphated glycosaminoglycan showing higher affinity for specific growth factors. Depending on the position and degree of the sulfur, five types of CS can be distinguished: CS-O, CS-A, CS-C, CS-D and CS-E. Each kind of CS has different biomedical properties. For instance, proteoglycan rich in CS-D and CS-E, which are oversulfated, promotes neural repair, unlike CS-A and CS-C [98].

Dextran is a hydrophilic polysaccharide composed of -1, 6-linked d-glucopyranose units. It can be obtained from sucrose and maltodextrins, using dextransucrase and dextrinise, respectively. In the biomedical field, dextran has been widely considered since it reduces inflammatory response and vascular thrombosis, and hinders ischemia-perfusion damage in organ transplantation [99]. It is also very interesting for tissue engineering because dextranase, a common enzyme in mammalian tissues, degrades it, and dextran’s structure can be modified, promoting enzymatic cross-linkable hydrogels.

Hyaluronic acid (HA), a negatively charged linear aminoglycoglycan, is composed of disaccharide unit repeats of β-d-glucuronic acid (GlcUA) and N-acetyl-β-d-glucosamine (GlcNAc), which are joined alternately by β-1,3 and β-1,4 glycosidic bonds. Its anionic appearance is due to the presence of carboxylate groups, which makes it highly hydrophilic [100]. Unlike the rest of the aminoglycans, HA is the only non-sulfated aminoglycan. However, it is sometimes preferable to incorporate sulfur, either to inhibit certain tumor-generating receptors such as HYAL-1 [101], to reduce degradation kinetics [102], or to increase interaction with growth factors [97].

### 3.2. Proteins

Collagen is one of the most abundant proteins produced by the human body and is found in the extracellular matrix of most connective tissues: skin, tendon, dentin, blood vessels, intestine, etc. The hallmark of collagen is a molecule that is composed of three polypeptide chains, each of which contains one or more regions characterized by the repeating amino acid motif (Gly-X-Y), where Gly is Glycine and X and Y can be any amino acid, but are often proline (Pro) and hydroxyproline (Hyp), making Gly-Pro-Hyp the most common triplet in collagen. Among the collagen family, the fibril-forming collagens (Types I, II, III, V, and XI) contribute about 90% of the total collagen content, with collagen Type I being the most frequently used and the most abundant collagen type in mammals [103]. Many studies work at acidic pH as collagen is soluble; however, this limits its applications as a bioink or injectable hydrogel because it is not cell-friendly. Recent reports have demonstrated that neutralized non-fibrillar Collagen I can be used as a bioink through a combination of enzymatic and thermal cross-linking.

Elastin fibers provide elasticity and resilience to many tissues such as arteries, lung, ligaments, tendons, skin, and elastic cartilage [104].These fibers are mainly composed of elastin, a protein obtained by the lysine-mediated cross-linking of tropoelastin. Although elastin plays a crucial role in tissue functionality, it cannot be used directly in tissue engineering applications due to its insolubility, such that elastin-based scaffolds are made of tropoelastin (elastin’s precursor), α-elastin, which is obtained by oxalic acid hydrolysis of elastin, and elastin-like polypeptides (ELP). ELPs are biopolymers constituted by repeated pentapeptide sequences of Val-Pro-Gly-Xaa-Pro, where Xaa can be any amino acid, except proline.

Gelatin is a denatured fibrous protein derived from collagen by partial thermal hydrolysis [105]. The denaturation of collagen involves the opening of the triple helix because of the destruction of the hydrogen bonds; however, the amino acid sequence remains constant. The chemical structure of gelatin consists of distinct polypeptide chains such as α-chains (one polymer/single chain), β-chains (two α-chains covalently cross-linked) and γ-chains (three α-chains covalently cross-linked) [106]. Due to its similar structure to collagen, gelatin is widely used in tissue engineering.

Silk is a protein that organizes in fibers, known as silk fibroin. Natural sources of this protein are spiders and silkworms (Bombyx mori) [107], the latter being the most common in tissue engineering as it mimics different tissues including cartilage, bone, skin, liver, nerve, etc. [108]. Silk fibroin is composed of repeating amino acid sequences of glycine, alanine, serine, and tyrosine residues—Gly-Ala-Gly-Ala-Gly-Ser, [Gly-Val]n-Gly-Ala, and [Gly-Val]n-Gly-Tyr [109]—making it a suitable candidate for enzymatic hydrogels.

### 3.3. Synthetic Polymers

Synthetic polymers have the opposite advantages and drawbacks to natural polymers. Polymers from industrial processes have tunable mechanical properties and less variability. However, their composition is very different from the native extracellular matrix [110], and they lack adhesion sequences for cell attachment. The most common polymer used in enzymatic cross-linking is poly(ethylene glycol) (PEG) due to its chemical structure and properties. The poly(ethylene glycol) monomeric unit is -[O-CH_2_-CH_2_]n-. When the chains are around 10 kDa, PEG receives the name of poly(ethylene oxide) (PEO) [7]. PEG is available in distinct structures such as branched, star and comb-like macromolecules [99]. Different enzymes have been used to synthesize PEG-based hydrogels.

## 4. Survey of Enzymes and Reactions

### 4.1. Transglutaminase

Transglutaminases (TGs) in mammals form a family of nine enzymes [111]. They catalyze the reaction between an acyl donor, such as the γ-carboxyamide group of a peptide-bound glutamine (Gln) residue, and an acyl acceptor, such as the ε-amino group of a peptide-bound lysine (Lys) residue, which results in the formation of an ε-(γ-glutamyl)lysine isopeptide bond [112,113,114] (see Table 2). Among mammalian TGs, tissue transglutaminase (tTG) or transglutaminase 2 (TG2) are the most used because they are involved in numerous physiological and pathological processes. TG2 structure is formed by four globular domains [111]. The core domain is responsible for the transamidation activity. The N-terminal β-sandwich domain has a binding site for fibronectin, and the two C-terminal β-barrel domains provide the ability to bind and hydrolyze guanosine/adenosine triphosphate. TG2 becomes activated in the presence of calcium (Ca^2+^). In order to have more accessible sources than mammalian TG, microbial transglutaminase (mTG) has been proposed and isolated. It is characterized by having a Ca^2+−^independent cross-linking activity, while being sensitive to other cations such as Cu^2+^, Zn^2+^, Pb^2+^ and Li^+^ [112].

Collagen is one of the main structural proteins of the extracellular tissue matrix that naturally contains the Gln and Lys residues for TG cross-linking [115,116,117,118,119]. In its native conformation, these residues are hidden within the triple helix and not accessible enough for TG cross-linking [111], which is why gelatin, the denatured version of collagen, is frequently used as an injectable hydrogel catalyzed by TG [120]. Gelatin with a higher bloom value shows higher mechanical strength and faster gelation than gelatin with lower values [121]. Differences in the mechanical properties of gels made from gelatin type-A and type-B have been described, type-A being the one that provides better mechanical properties [85].

As most polymers used to prepare hydrogels do not have the glutamine and lysine residues, some previous structural modifications are necessary in the precursor macromolecules to graft them. For instance, pure hyaluronan-based hydrogels can be enzymatically cross-linked using transglutaminase by mixing Gln and Lys HA conjugates [122]. H-Gly-Lys dipeptide can be used for functionalizing hyaluronic acid in order to mix it with gelatin and cross-link the mixture with TG [123]. Hu and Messersmith modified PEG with peptides containing lysine and glutamine, and used transglutaminase from guinea pig liver for cross-linking [124]. Mixtures of HA and PEG can be prepared after modifying them for TG cross-linking. This is the case of the HA-Gln conjugates that are mixed with PEG by previously preparing an eight-arm PEG macromer terminated with the lysine donor peptide [125]. Interesting bone marrow organoids of PEG-HA hydrogels have been prepared by TG cross-linking an eight-arm PEG-Gln or an eight-arm PEG-MMP-sensitive-Lys, with a HA-Gln or a HA-MMP-sensitive-Lys in the presence of Ca^2+^ [126].

Elastin-like polypeptides (ELP) can be used to form TG cross-linked hydrogels. McHale et al. designed lysine and glutamine containing ELP by substituting the residue in position X of the ELP repeat sequence, and succeeded in the encapsulation of chondrocytes after human recombinant tissue TG cross-linking [127]. Modified ELP can be combined with collagen via mTG cross-linking to regulate the mechanical properties of the resulting hydrogels. The published results have shown that the increase in the collagen content decreased the storage modulus due to a lower polymer compaction [128]. TG cross-linking has been extended to other proteins such as Bambara groundnut protein isolate [129].

### 4.2. Phosphopantetheinyl Transferase

Transferases are composed of large multifunctional polypeptides that contain all the catalytic components necessary for the synthesis of long-chain fatty acids [79]. As the main reaction mechanism, phosphopantetheinyl transferase (PPTase) can transfer the 4-phosphopantetheine moiety of coenzyme A (CoA) to a serine residue. With this approach, it can modify and covalently bond to a carrier protein. This function plays an important role in the biosynthesis of natural products [130,131]. Even though the enzyme has been occasionally used to form injectable hydrogels, it is a good alternative to transglutaminase because it can be obtained with a high expression yield. The reaction usually takes place at 37 °C in the presence of Mg^2+^ and a neutral pH. Mosiewicz et al. pioneered the use of this enzyme in the preparation of hydrogels by combining a CoA-functionalized poly(ethylene glycol) (PEG) and an engineered apo-acyl carrier protein (ACP) dimer with two phosphopante-theinylation sites. For hydrogel forming, they used stoichiometrically balanced liquid ACP_ACP and 8-PEG-CoA precursors. The liquid precursors were mixed, and after adding PPTase (1.0 mM) in a 5% (*w*/*v*) precursor solution, the gelation process took place within 15 min. With this composition, the elastic modulus reached was 2.3 kPa at a frequency of 1 Hz. The 3D in vitro cell assay showed satisfying results. The authors concluded that hydrogels could be improved by replacing the ACP_ACP linker protein by short peptide analogues for faster cross-linking [131]. Perhaps due to the complexity of the system, the strategy of using PPTase to prepare enzymatically gellable hydrogels has not been further explored in other polymers.

### 4.3. Tyrosinase

Tyrosinase is an enzyme with copper in the active center that can be widely found in our living system [132]. It has two copper ions (CuA and CuB) in the active site that are highly coordinated with six histidine residues. The mechanism of tyrosinase reaction consists of two consecutive oxidations. Firstly, the copper ions in the active site of the deoxy-tyrosinase (an oxidated state of the tyrosinase) bind two oxygen atoms, turning into oxy-tyrosinase. This oxy-tyrosinase produces the second oxidation that can be classified into two types, depending on the substrate. If the substrate is the hydroxyl group in phenol, the oxidation mechanism is called monooxygenas, and the products are catechol and ortho-quinones. If the substrate contains two hydroxyl groups in cathecol, the oxidation mechanism is called oxidase, and the product is only ortho-quinone. As ortho-quinone has high electrophilicity, it can participate in a variety of reactions such as Schiff-base reaction, Michel-type addition, and coupling reactions [79,133]. In Schiff-base reaction, the carbonyl group (C=O) of the ortho-quinone reacts with primary amines producing ortho-quinone imines containing the C=N bond. In the Michel-type addition, the ortho-quinone can react with a primary amine or a thiol group, which can be attached to the meta-position and ortho-position. In the coupling reaction the ortho-quinone interacts with neighboring catechol groups, forming a covalent bond.

Tyrosinase oxidations occur using phenolic moieties and oxygen, without any other cofactor, and this means that it occurs in mild conditions that are compatible with cells. Hydrogel precursor components and enzymes are dissolved in a phosphate-buffered saline solution, freshly prepared, and enzymatic reaction takes place at 37 °C.

Pioneer studies selected phenol-containing proteins of the extracellular matrix of tissues such as collagen or gelatin for their cross-linking, because they can be used without any previous chemical modification. As they also contain primary amine and thiol groups in their amino acids, their chains can react with the tyrosinase-mediated oxidated phenols of other chains [134]. A more efficient strategy has been the mixture of gelatin with other hydrogels that are rich in free amine groups such as chitosan [135] or silk fibroin (this second also contains tyrosine residues) [136].

With the evolution of the field, it has been demonstrated that previous polymer structure modifications to incorporate phenol or cathecol groups to polymers of interest in tissue engineering can allow their cross-linking with tyrosinase. A common modification consists of the reaction of carboxylic acid groups of hydrogels, COOH, with amine reactants of tyramine or tyrosine, through an amide bond, leading to phenolic polymers. The same reaction is usually performed with dopamine to obtain cathecolic polymers. This strategy was followed by Jin et al. to synthesize in situ hydrogels based on the tyrosinase-mediated cross-linking of chondroitin sulfate-tyramine conjugates. They demonstrated that the gelation decreased at higher tyrosinase concentrations [137]. Similarly, Kim et al. mixed tyramine-modified hyaluronic acid with gelatin to mimic the composition of the extracellular matrix, using the common cross-linking reaction with tyrosinase for oxidizing the phenol substrates of the hyaluronic acid conjugate and non-modified gelatin. Because gelatin also contains amine and thiol groups, cross-linking can also happen via Michael-type addition or Schiff-base reaction. They showed that this gel can be used for injecting and spraying [138]. Hydrogels based on gelatin previously modified with hydroxyphenyl propionic acid to provide phenol groups have also been proposed for dual cross-linking by a combination of two enzymes, horseradish peroxidase and tyrosinase [139]. When the tyrosinase concentration is increased, the gelation time also increases, and the storage modulus is reduced due to the competitive reaction of the enzymes.

### 4.4. Horseradish Peroxidase

Horseradish peroxidase (HRP) is an oxidoreductase that catalyzes a variety of oxidative reactions. Various studies found that horseradish contains several peroxidase isoenzymes [140,141,142]; however, the fundamentals of HRP are based on isoenzyme C. HRP isoenzyme C is formed by 308 amino acid residues, four disulfide bridges between cysteine residues as well as a heme group (iron(III) protoporphyrin IX), and two calcium atoms.

The HRP cross-linking mechanism starts with the interaction between the ferric state of HRP (Fe(III)) and H_2_O_2_. In this step, H_2_O_2_ is oxidized to form water and HRP is reduced to produce Compound I, which is a high-oxidation-state intermediate with a cation radical that interacts with a phenol group as a reducing substrate. Compound I is transformed into Compound II due to one-electron reduction, followed by a second one-electron reduction that returns HRP to its neutral stage. On the other hand, the phenol radicals interact among themselves forming C-C or C-O cross-linking, which results in the 3D network.

HRP thus needs H_2_O_2_ to become reactive in order to interact with the polymer. This polymer should have functional groups, such as aromatic phenols, amines, and phenolic acids, to play the role of a reducing substrate. These functional groups are commonly introduced by tyramine. Feng Chen et al. developed an injectable hydrogel composed of carboxymethyl pullulan-tyramine and chondroitin sulfate-tyramine conjugates [143]. Moreira Teixeira et al. developed interesting in situ gelling dextran-tyramine hydrogels for cartilage regeneration [144]. Poveda-Reyes et al. prepared mixtures of gelatin-tyramine and hyaluronan-tyramine derivates with promising results for soft tissue regeneration [145]. Although tyramine has been used widely, chemical reactions using phenylalanine, tyrosinase, hydroxyphenyl propionic acid and 4-hydroxyphenyl acetic acid also allow phenol incorporation [146]. For instance, R. Jin et al. modified chitosan’s structure by grafting glycol acid and phloretic acid groups to generate phenol groups [147].

On the other hand, the higher the amount of H_2_O_2_, greater the stiffness of the resulting hydrogel—up to a certain limit. Excessive H_2_O_2_ results in two different types of HRP inactive compounds known as Compounds III and IV. While Compound III can reduce slowly and transform into the neutral HRP, Compound IV is irreversible, so that less stiff hydrogels can be obtained by using a H_2_O_2_ concentration above the optimal, since the cross-linking reaction is inhibited [30]. Methods using HRP without adding hydrogen peroxide have also been developed [86,87,148]. Sakai et al. made tried to use HRP without adding hydrogen peroxide, using glucose oxidase (GOX), which oxidized glucose to hydrogen peroxide, for generating hydrogen peroxide for the HRP reaction. In this way they ensured that the hydrogen peroxide is gradually supplied as the enzymatic reaction progresses [149]. Gantumur et al. showed that it is also possible to use HRP-mediated cross-linking without additional hydrogen peroxide and without glucose oxidase by only adding glucose [150].

Other important parameters such as pH, medium composition and temperature should also be considered. HRP has polar amino acids on the surface that can be protonated or deprotonated according to the medium’s pH. The optimal pH range is between 5 and 9, near the physiological pH, at which HRP has its maximum activity. The composition of the medium also plays a crucial role. Water is the most prominent candidate, since it gives flexibility to the active site. Organic solvents are not commonly considered as they promote enzyme inactivation by modifying their structure. As regards temperature, a higher temperature, higher reaction and velocity also lead to the loss of interaction forces, reducing or even de-activating enzyme activity [142].

### 4.5. Sortase

Sortases (Srt) are bacterial enzymes responsible for the covalent binding of specific proteins through the transpeptidation reaction. Different sortase isoforms have been analyzed [151,152], with Sortase A (SrtA), known as the “housekeeping sortase”, being the most prominent candidate. SrtA has been reported to facilitate covalent modifications of proteins under mild conditions [153]. SrtA is composed of an uneven eight-stranded β barrel connected by random coil loops. The active site is at the end of the barrel and consists of the catalytic residues histidine120, cysteine184 and arginine197. The specific proteins that react with SrtA are characterized by having three crucial features at their carboxy terminal: a hydrophobic region, charged residues and an LPXTG motif (leucine, proline, X, threonine, and glycine, being X any amino acid). The hydrophobic region and charged residues allow the recognition of the LPXTG sequence by SrtA followed by a transpeptidation reaction. Cysteine184 of SrtA reacts with the carbonyl carbon atom of the peptide bond between threonine and glycine, forming a thioester intermediate. The terminal amino group of an oligoglycine-functionalized compound on the thioester of the acyl-enzyme intermediate generates an amide bond between the SrtA-bound target protein moiety and the coupling partner. The product of the reaction is released and the enzyme can start a new reaction, since the active site is available [153].

This catalytic reaction was undertaken by Arkenberg et al. to form PEG hydrogels. PEG was modified to eight-arm PEG-norbornene (PEG8NB), and peptides bearing a cysteine residue were added via thiol-norbornene photoclick chemistry. The results show that a higher SrtA concentration reduced gelation time and increased the storage modulus [154]. Matthew R.et al. produced peptide linkers containing the SrtA-specific sequence to obtain the controllable cross-linking of PEG-peptide hydrogels. They also designed a peptide cross-linker attached to the SrtA substrates to analyze the reversibility of the hydrogel stiffness [155]. Natural polymers such as hyaluronan have also been used in sortase cross-linking. Broguiere et al. used hyaluronic acid modified by SrtA substrate peptides to study the cross-linking kinetics, stability, and cytocompatibility of the hydrogel [156]. Broguiere et al. compared gels made from modified hyaluronic acid (HA) and modified four-arm-polyethylene glycol (PEG) incorporating the amino acid motif. Hydrogels containing HA-LPXTG required less gelation time than PEG-LPXTG for the same amount of SrtA.

**Table 2 gels-09-00230-t002:** Summary of the most common enzymatic cross-linking mechanisms for bioinks and injectable hydrogels.

Enzymatic Cross-Linking	Polymers	References
HRP/H_2_O_2_ and Tyrosinase 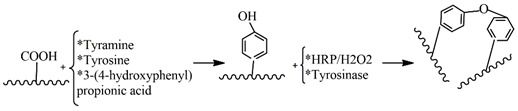	Chitosan	[135]
	Chondroitin Sulfate	[137,143]
	Dextran	[144]
	Hyaluronic Acid	[138,145]
	Collagen	[134]
	Gelatin	[135,138,139,145]
	Silk	[136]
Transglutaminase 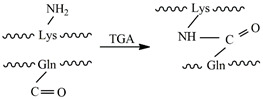	Collagen	[115,116,117,118,119,123]
	Gelatin	[85,120,121]
	Hyaluronic Acid	[122,125,126]
	PEG	[124,126,157]
	Elastin	[128]
Alkaline Phosphatase 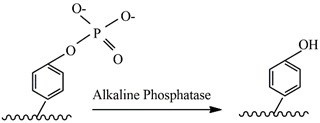	Collagen	[158]
	Synthetic	[159,160]
Sortase 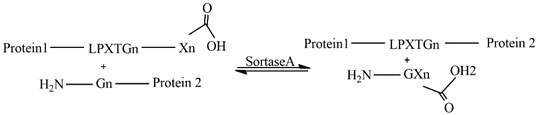	Hyaluronic Acid	[156]
	PEG	[154,154,155,161]
Phosphopantetheinyl transferase 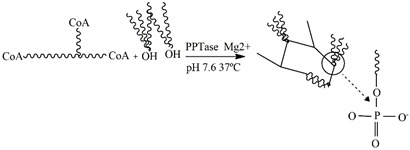	PEG	[131,162]

### 4.6. Alkaline Phosphatase

Alkaline phosphatase (ALP) is a low-specificity metalloenzyme responsible for the hydrolysis of a wide variety of phosphate esters. Different substrates can be catalyzed by ALP such as p-nitrophenyl phosphate, β-glycerophosphate, phenylphosphate and phenolphthalein [163]. Its structure consists of three metals ions in the active site, two Zn^2+^, M1 and M2, and one Mg^2+^, known as M3. M1 is coordinated into three histidine residues (His-331, 372 and 412) and is essential for enzyme activity. M2 is attached to Asp-369, His-370 and Asp-51, the latter bridging M2 to M3 with a carboxy oxygen, and the protein ligands to M3 are Asp-51, Asp-153, Thr-155 and Glu-322 [164]. A key factor in ALP activity is the presence of a serine residue, Ser-102, whose oxygen atom works as a nucleophile, activating serine by hydrogen-bond formation with other acid-base groups in the active site. An acyl- or phosphoryl-enzyme intermediate is formed, which is then hydrolyzed with the re-addition of a proton to the serine oxygen. pH plays a crucial role in the reaction, since the lability of the phosphoryl-enzyme’s intermediate and catalytic activity increases with it, depending on the state of ionization of an active-site group [165]. ALP has thus been widely studied since it is a glycoprotein that catalyzes the hydrolysis of different phosphate monoesters in mild conditions.

Hai et al. used ALP for the hydrogelation of Fmoc-Phe-Phe-Tyr(H_2_PO_3_)-OH (FmocHOH). After 15 min, they observed gel forming in the mixture from FmocHOH and Na_2_CO_3_ with ALP. It was initially a turbid gel, but after 15 min it became clear [88]. Criado-Gonzalez et al. used ALP to enhance the visco-elastic properties of a PEG hydrogel, which was cross-linked by thiol–ene click chemistry. They physically trapped the ALP in the gel during gel formation and created an enzymatically active hydrogel [159]. Chen et al. worked on a photo-cross-linked hydrogel based on Collagen (Col) and ALP modified with methyl acryl groups. ALP’s catalytic function is used to mineralize the hydrogel. To reach this result, vinylphosphonic acid and glycerol phosphate calcium salt hydrate (CaGP) were also added. ALP catalyzes the hydrolysis of CaGP. The C-PO_4_^3−^ group of vinylphosphonic acid promotes binding between calcium phosphate ion clusters and Col matrix molecules, which starts mineralization. This simulates the physiological mineralization process and the bone development stiffening process. They performed the first printing experiments using the hydrogel as a bioink, and reported high viability [158]. Yu et al. used ALP with the hydrogelator Nap-Phe-Phe-Tyr(H_2_PO_3_)-OH (NTP) as a drug delivery hydrogel for methylprednisolone sodium phosphate (MP). ALP catalyzes the hydrolysis of NTP to Nap-Phe-Phe-Tyr-OH (NT). They compared one gel without MP with another containing MP. The gel containing MP showed thicker nanofibers and a higher storage modulus [160].

## 5. Application as Injectable Systems in Tissue Engineering

Injectable systems are important in tissue engineering and have been widely studied. The application of these systems consists of regenerating a patient’s tissues and organs or acting as a vehicle for transporting drugs by injecting a specific solution into the damaged area (Figure 5). The injected solution is prepared according to the tissue selected, choosing suitable polymers, biomolecules, cells and cross-linking mechanisms. Once the solution is inside the body, the in situ 3D hydrogel is formed and fits perfectly in the shape of the defect. Injectable hydrogels have the ability to self-conform to the function of the affected area. The encapsulated cells adhere to or interact with the polymer chains, proliferate, and differentiate to the adequate phenotype, obtaining an optimal cell–matrix interaction. As the cells secrete a new extracellular matrix to regenerate the affected area, the scaffold degrades via enzymatic action. Tissue is recovered and the hydrogel is eliminated without affecting other organs. Non-invasive injectable hydrogels are thus considered prominent candidates for clinical applications, since they can reduce surgical interventions and the procedure time, and avoid the risk of infections. Apart from considering all the characteristics of the hydrogels cited in Section 1, certain parameters are crucial. To be injectable, a hydrogel must be liquid before and during injection and become solid inside the body. This means that the surgeon must have enough time to inject the solution and the hydrogel must be sufficiently robust to promote cell interactions, so that gelation time, viscosity, biodegradability, degradation kinetics, stability, and mechanical properties are crucial parameters that can be controlled by modifying enzymatic cross-linking reactants. For instance, Yajie Zhang et al. developed an injectable hydrogel system via the HRP/H_2_O_2_ reaction by mixing hyaluronic acid–tyramine (HA-TA) and chondroitin–tyramine (CS-TA) derivates. The results show that the higher the concentrations of HRP and H_2_O_2_, the faster the hydrogels formed and the harder they became. The optimal concentration was 1.5 units/mL for HRP and 4 mM for H_2_O_2_, obtaining a gelation time of 15 s and a storage modulus of 2.8 kPa [166]. McHale et al. synthesized elastin-like polypeptide (ELP) hydrogels through transglutaminase cross-linking for cartilage repair because of their potential use in promoting chondrogenesis and controlling the mechanical properties. The results show the maintenance of chondrocyte phenotype for cells and the increased mechanical integrity of the cross-linked hydrogels, from 0.28 to 1.7 kPa during a 4-week culture period, suggesting a restructuring of the ELP matrix by deposition of cartilage extracellular matrix components [127]. Wei et al. designed an injectable hydrogel using dual cross-linking: the enzymatic and Diels-Alder reaction. Poly(γ-glutamic acid) (PGA)-based hydrogels were synthesized through furfurylamine and tyramine(Tyr)-modified PGA and two cross-linkers: dimaleimide poly(ethylene glycol) and H_2_O_2_. The results show that mechanical properties, swelling ratio, pore size and degradation behavior could be easily controlled by modifying the molar ratios of H_2_O_2_/Tyr and furan/maleimide. Hydrogels also exhibited sustained release behavior encapsulating bovine serum albumin [167]. Kuo et al. developed collagen injectable hydrogels as support for the formation of vascularized tissue grafts by human blood-derived endothelial colony-forming cells (ECFCs) and bone marrow-derived mesenchymal stem cells (MSC) in vivo. The collagen backbone has previously been modified with phenol groups to produce HRP/H_2_O_2_ cross-linking. The solution mix of the polymer, cells and cross-linkers was injected into the subcutaneous space or abdominal muscle defect of an immunodeficient mouse. The authors concluded (1) extensive human ECFC-lined vascular networks can be generated within 7 days, (2) improvement of the long-term differentiation of the MSCs into mineralized osteoblasts was achieved, (3) an increase in the adipocytes within the vascularized hydrogel in a mouse occurred after one month of implantation, and (4) changing the mechanical properties and proteolytic degradability was possible, thus improving the system [168]. Shujie Hou et al. demonstrated that gelatin injectable hydrogels cross-linked by transglutaminase maintained their mechanical properties at 37 °C, and were also degraded by collagenases and gelatinases. The results show that pore size promoted cell adhesion, proliferation and migration, and cell culture studies concluded that they were suitable platforms for the encapsulation of cells and growth factors [169].

## 6. Bioprinting Applications

Bioprinting is a promising biofabrication method that consists of the automatic construction of 3D tissue-like structures by deposition of a bioink with a pre-programmed geometry in a device called a bioprinter. The bioink is a viscous liquid formed by hydrogel precursor materials, living cells and biomolecules. When the bioink is deposited in the bioprinter plate, hydrogel cross-linking takes place and the structure keeps its shape while the cells and biomolecules remain trapped. Bioink deposition is synchronized with the hydrogel cross-linking and the monitored movement of the printer head [171]. To produce 3D structures, the printing process is performed layer-by-layer. Contemporary bioprinters can build complex structures using different heads for the deposition of multiple materials and cells [172]. Hydrogel synthesis through the bioprinter is illustrated in Figure 6.

The most widely used bioprinting techniques are ink jet bioprinting, laser bioprinting and extrusion bioprinting [173]. Ink jet bioprinting consists of depositing picoliter-range ink droplets generated in the print head by a thermal or piezoelectric actuator. To avoid nozzle clogging, low-viscosity materials must be used [174]. The advantage of this technique is its low cost, while its main disadvantage is the low printing resolution. Laser bioprinting consists of transferring the bioink from an upper donor substrate to a lower receiving substrate. Initially, the bioink is in contact with a sacrificial material that is vaporized by the laser, generating the bioink droplet to be transferred to the substrate. This technique does not require a dispensing nozzle, and avoids shear stress in the cells and nozzle clogging problems. Its advantage is that it allows the printing of viscous bioinks with high cell densities and good resolution, which can be up to 10 µm. However, its complexity and high cost have hindered its use in tissue engineering [173]. Most commercial bioprinters use extrusion to dispense the bioink as a continuous filament or as droplets using a piston, a screw or by pneumatic pressure [174]. To generate droplets in extrusion-based bioprinters, a microvalve is generally used. The upper head containing the bioink automatically moves in three directions to build the scaffold. The technique can deposit highly viable viscosity bioinks with high cell densities. Although there is a risk of nozzle clogging, it is usually lower than in ink jet technology, although it has a lower resolution than laser bioprinting [172,174].

**Figure 6 gels-09-00230-f006:**
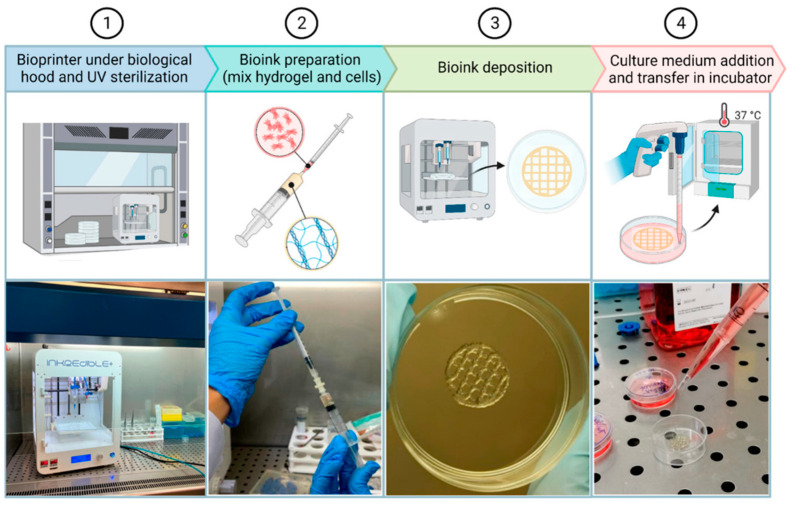
Three-dimensional bioprinting procedure. It starts with (1) the positioning of the bioprinter under a biological hood and UV light sterilization. (2) Next, bioink is prepared by mixing cells and hydrogel, and (3) printing finally takes place. (4) Culture medium is added, and 3D structures are placed inside the incubator. (reprinted with permission from [175]).

### 6.1. Viscosity-Rheological Parameters

The bioink viscosity should be such that the applied pressure does not damage the cells and provides a stable system after bioprinting. Shear thinning materials are used as bioinks since their viscosity decreases under shear stress. The pressure applied in the nozzle reduces viscosity and facilitates bioink deposition. In this step, the relation between the applied pressure and cell viability must be considered. When the shear rate is removed after exiting the orifice, the viscosity increases or remains constant [176]. The resulting viscosity of the bioink must provide a reliable and stable architecture for the 3D structure. Low-viscosity bioinks produce better cell viability than high-viscosity bioinks, as the pressure applied is lower, even though a stable structure is not obtained, while high-viscosity bioinks obtain robust structures but present the problem of applying high pressures and thus cell damage. Viscosity is therefore crucial before and after the bioprinting process, so that a balanced viscosity must be obtained. This is affected by different parameters: the concentration and molecular weight of the polymer, the polydispersity index, cell density and distribution, and processing temperature.

The higher the polymer’s concentration and molecular weight, the higher the viscosity due to an increase in chain entanglements. For instance, Ouyang et al. demonstrated an increase in gelatin viscosity from (0.57 ± 0.04) Pa s to (1.84 ± 0.16) Pa s when going from a concentration of 5% to 10% [177]. The polydispersity index (PDI) is related to both parameters, and indicates the variability of the polymer chain size. Low PDI means that the polymer chains are similar, resulting in consistent and uniform mechanical properties [178], while a polymer with a high PDI requires a more thorough study of its viscosity.

It has been shown that the cells modify a bioink’s rheological properties [176]. The type, concentration and distribution of the cells can increase, reduce or keep the viscosity constant [42]. Martorana et al. worked with two types of cell line models, murine preostoblastic cells (MC3T3-E1) and human colon tumour cells (HCT-116), at different concentrations, and found the presence of HCT-116 increased the bioink viscosity up to 1 M/mL, after which it decreased. For MC3T3-E1, only the smallest density used showed a slight increase compared with the bioink without cells. The highest densities resulted in a decrease in viscosity [179]. Different authors have found reduced viscosity [180,181], since the cells could act as a physical hindrance between the different regions of the bioink or limit contact between the reacting groups. It has also been shown that cells can increase the rheological properties due to their own characteristic mechanical properties [182]. Viscosity can vary before and after hydrogel formation. Diamantides et al. demonstrated that for collagen bioinks, the viscosity behavior was different before and after gelation with the incorporation of primary chondrocytes. Before gelation, the mechanical properties increased with cell density, but after gelation, increasing in cell density reduced the mechanical properties [183].

Bioprinting process parameters, such as temperature, can also alter the bioink viscosity. Martin et al. showed that for a bioink containing 20% PLGA-PEG-PLGA, viscosity decreases with temperature, whereas it increases for 20% Poloxamer 407. On the other hand, viscosity remains almost constant with temperature for a 12% silk-elastinlike polymer solution [184]. Ouyang et al. demonstrated that the viscosity of 7.5% gelatin increased sharply at temperatures lower than 25 °C, while alginate, which was not temperature-responsive, did not vary the viscosity [177].

### 6.2. Nozzle/Needle

Cell survival depends not only on the pressure applied, but also on the diameter and type of needle. Cell viability decreases as the nozzle diameter decreases [185], and cells are less affected by the shear imposed by conical rather than cylindrical nozzles [181]. Although peak shear stress is obtained in conical needles, their exposure time is shorter than for cylindrical needles. The latter generate a lower peak shear rate with prolonged passage length, causing cell damage [42]. Billiet et al. evaluated the influence of needle type (conical vs. cylindrical), needle internal diameter and processing pressure on the viability of HepG2 cells in a gelatin methacrylamide bioink. For both types of needles, a higher inlet pressures resulted in reduced cell survival. However, at the same low inlet pressure (<1 bar), conical needles obtained the highest viability (>97%). This result was explained by heat maps, which showed the distribution of the shear stress and outlet pressure through the needle. Although in the conical needles the cells were subjected to the greatest stress, they were at the end of the path (<1 mm). In contrast, the stress in cylindrical needles was lower but longer-lasting (>16 mm), damaging the cells [186]. In relation to the bioink outlet, the bioink must have optimum surface tension to avoid sticking to the nozzle, regardless of the nozzle type [187].

### 6.3. Influence of Cross-Linking Time

The extruded bioink needs to cross-link in an adequate time before spreading to obtain constructs with the desired shape without compromising printing resolution, a concept that has been called shape fidelity [188]. Enzymatic gelation time can be controlled by the enzyme concentration, as demonstrated by Zhou et al. in a gelatin bioink cross-linked by mTG, in which gelation time decreased when the enzyme concentration increased [189].

In the case of HRP-catalyzed hydrogels, the cross-linking reaction is so fast that the direct addition of H_2_O_2_ in the bioink is a challenge. Cells do not survive for long in H_2_O_2_ solutions since they are cytotoxic. This problem has been solved by using two printing nozzles, one containing an ink with the minimum amount of H_2_O_2_ to cross-link the hydrogel and keep the shape of the construct, and the other containing the polymer, the enzyme, and the cells [190]. Another method of supplying hydrogen peroxide consists of vaporizing hydrogen peroxide solution to obtain a predictable gelation and good printing resolution [191]. A recent publication showed that cross-linking time can be delayed by adding a certain amount of glucose to the bioink to indirectly produce H_2_O_2_ by the reaction of HRP and glucose [188]. In this case, to ensure the shape fidelity of the printed construct during cross-linking, they added a dispersion of cellulose fibers to increase the bioink’s viscosity. Long printing procedures of up to 20 min were performed by adding glucose. The authors suggest that longer printing times can be achieved by reducing the amount of HRP and glucose. Longer cross-linking times of about 1 h were considered optimal for a composite bioink based on silk fibroin and gelatin enzymatically cross-linked by mushroom tyrosinase (at 500 units) and sonication [136].

### 6.4. Swelling Properties

After the printing process, a balance between the swelling, stability and degradation of the printed scaffold is important to its successful use in tissue engineering. The swelling enables oxygen and nutrient transport. As highly cross-linked structures have reduced swelling, the consequent low diffusion can compromise cell viability. However, shape fidelity is better maintained in highly cross-linked structures than in lower cross-linked ones. Cross-linking density should be enough to provide mechanical stability, and at the same time provide nutrients to the cells and allow them to move and proliferate within the hydrogel [172].

## 7. Conclusions and Future Perspectives

Hydrogels, which are biomimetic platforms for different applications in tissue engineering, can be synthesized by physical or chemical cross-linking. Enzymatic cross-linking is one of the best chemical reactions, as it does not use exogenous reagents but the subject’s own enzymes work under mild conditions, is a spontaneous reaction that provides rapid gelation, and is a specific controlled reaction. However, despite the enzymes’ great advantages, there are still challenges to be overcome. These mainly relate to the difficulty of producing and obtaining the enzymes, the prior modification of the natural and synthetic polymers to be cross-linked, and the limited mechanical properties of the hydrogels produced. Future development directions will be focused on: (1) combining different types of enzymes to enhance the mechanical properties of the native tissue, (2) using simple chemistry in the previous modification of the polymers, where for example for peroxidase and tyrosinase a phenol group is introduced into the polymer backbone, (3) the development of more recombinant enzyme types to include a wide variety of mechanisms, and (4) considering enzymatic cross-linking for the next generation of biomaterials for tissue engineering applications for its exceptional control of hydrogel formation, providing a step towards higher complexity, non-cytotoxicity and non-invasiveness. In this context, within the different applications, tissue regeneration by injectable hydrogels and fabrication of an automated 3D system from bioinks by bioprinting will play a fundamental role in tissue engineering. A greater number of hydrogels will be developed in the near future for biomedical applications in personalized medicine, and this will advance the tissue engineering field.

## Figures and Tables

**Figure 1 gels-09-00230-f001:**
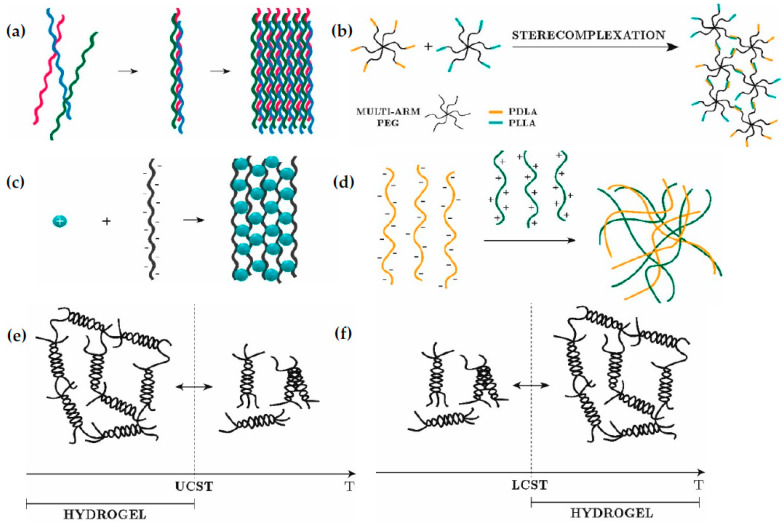
Physical cross-linking. (**a**) Self-assembly. (**b**) Stereo-complexation. (**c**) Ionic-chelation. (**d**) Ionic–electrostatic interaction. (**e**) Thermal condensation UCST. (**f**) Thermal condensation LCST.

**Figure 2 gels-09-00230-f002:**
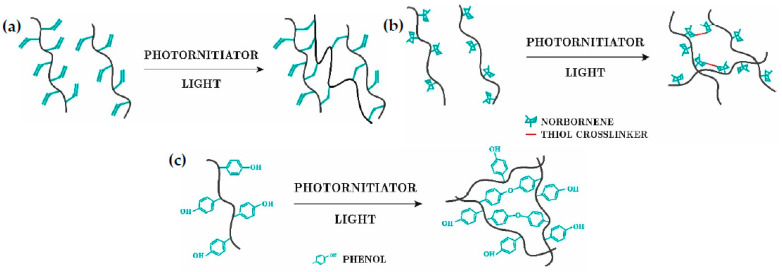
Photo-induced cross-linking. (**a**) Free-radical chain. (**b**) Thiol-ene photo. (**c**) Photomediated redox reaction.

**Figure 3 gels-09-00230-f003:**
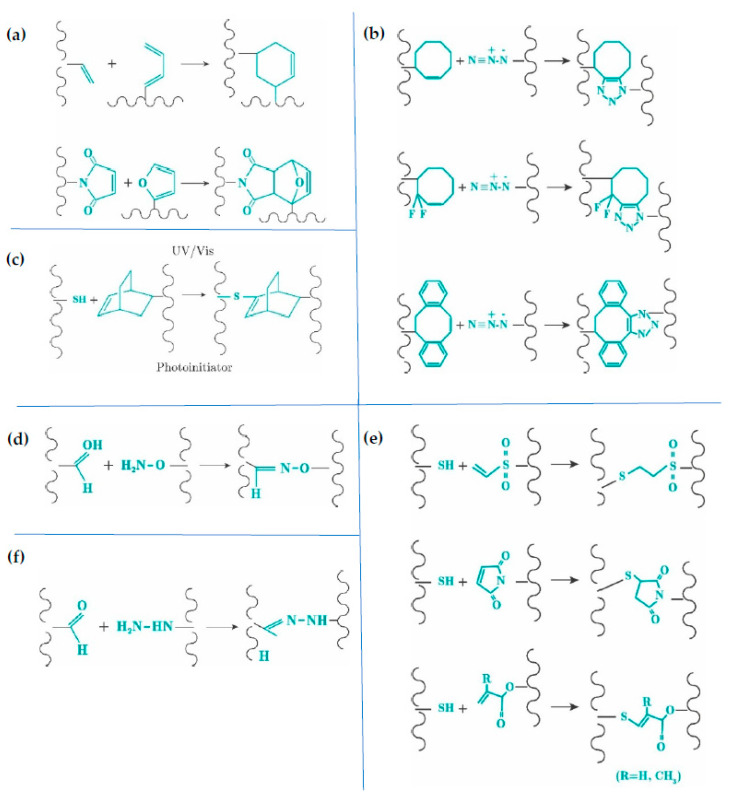
Click chemistry. (**a**) Diels-Alder reactions. (**b**) Strain-promoted azide-alkyne cycloaddition (SP-ACC). (**c**) Thiol-ene click chemistry. (**d**) Oxime reaction. (**e**) Thiol-Michael pseudo-click reaction. (**f**) Aldehyde-hydrazide pseudo-click.

**Figure 4 gels-09-00230-f004:**
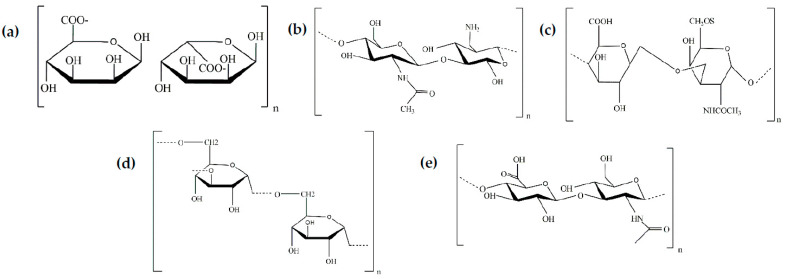
Monomeric unit of polysaccharides. (**a**) Alginate. (**b**) Chitosan. (**c**) Chondroitin sulfate-C. (**d**) Dextran. (**e**) Hyaluronic acid.

**Figure 5 gels-09-00230-f005:**
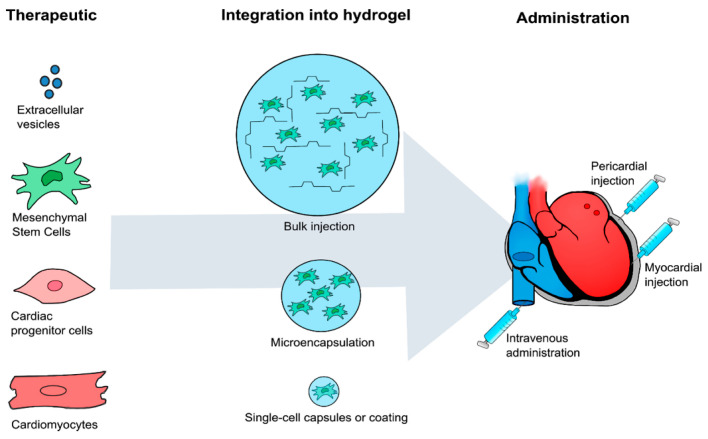
Strategies for cardiac diseases using injectable hydrogels. The therapeutics have been integrated into hydrogels and delivered by bulk injection, microencapsulation, and single-cell capsules/coating. Combinational therapy has primarily been administered through intramyocardial injections, but intravenous and pericardial injections have also been reported (reprinted with permission from [170]).

## Data Availability

Not applicable.

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
