# Peer review of "Research Progress in Enzymatically Cross-Linked Hydrogels as Injectable Systems for Bioprinting and Tissue Engineering"

_gels, 2023, doi:10.3390/gels9030230_

Round 1

Reviewer 1 Report

Dear Authors

This manuscript written by Raquel Naranjo-Alcazar et al. is a research on. " Research Progress in Enzymatically Cross-linked Hydrogels as Injectable Systems for Bioprinting and Tissue Engineering ”. the study is unacceptable in its current form, manuscript needs to majör revision

Reviewer 2 Report

Hydrogels have been developed for different biomedical applications, such as tissue engineering, drug delivery, and in vitro culture platforms, due to their excellent properties such as biocompatibility and low toxicity. Among them, enzymatic crosslinked hydrogels are able to form gels in situ when injected into tissues, which provides conditions for minimally invasive surgery. And because of their good biocompatibility, it will not cause any harm to cells during the crosslinking process, which is in contrast to the hydrogels formed by chemically induced crosslinking. In this review, the authors outlined enzymatic crosslinking mechanism of hydrogels. And then they summarized the application of enzyme crosslinking mechanism in natural and synthetic hydrogels. Their applications in bioprinting and tissue engineering are also analyzed. This work is effective, and can be recommended to be accepted after finishing the following minor revisions:

1. In section “5. Application in tissue engineering as injectable systems”, the author only lists a few examples. This part should be enriched.

2. Appropriate figures should be added in section “5. Application in tissue engineering as injectable systems” and “6. Application in bioprinting” to make it easier for others to understand the content of their application 

3. In section “7. Conclusions and Future Perspectives”, whether the author can supplement the existing problems of enzymatically crosslinked hydrogels, as well as the future development direction?

4. In the references, some journal name formats need to be modified, for example: ref. 7913, the name “Advanced MaterialsAccounts of chemical research” needs to be abbreviated.

5. In the abstract, page 1, line 14, change “Enzymatically” to “Enzymatic”.

6. In the abstract, page 1, line 19, change “opens also” to “also opens”.

7. In the main text, page 4, line 193, change “do” to “does”.

8. In the main text, page 5, line 217, change “chemistry” to “chemical”.

9. In the main text, page 16, line 700, change “three-dimensional” to “3D”.

10. In the main text, please keep the formats of "cross-linking" and "crosslinking" consistently through all the paper, for example: page 5, line 247, crosslinking; page 1, line 40, cross-linking.

Reviewer 3 Report

The authors have collected and summarized cross-linking mechanism of hydrogels and further surveyed enzymatic cross-linking mechanism applied to both natural and synthetic hydrogels. The authors have also provided the applications of enzymatically cross-linked hydrogels in bioprinting and tissue engineering. Overall, this manuscript is a well-written and well-organized review paper. Therefore, I would like to recommend this work to publish in Gels. Below are some comments for the authors.

1. For Table 1, the related references should be added in Table 1.

2. For the section “7. Conclusions and Future Perspectives”, this review would be more impressive if the author could provide the challenge for research progress in enzymatically cross-linked hydrogels as injectable systems for bioprinting and tissue engineering.

Reviewer 4 Report

In this review article, the authors present the progress in research on enzymatically cross-linked hydrogels as 2 injectable systems for bioprinting and tissue engineering. Enzymatic crosslinking is one of the best as it does not use exogenous reagents, it works under mild conditions, it is a spontaneous reaction that provides rapid gelation, and it is a specific controlled reaction. As the authors claim, a greater number of hydrogels will be developed for biomedical applications for personalized medicine, bringing the field of tissue engineering to society in the future. In this place, bioprinting plays a key role since it will allow obtaining hydrogels on an industrial scale. However, due to poor presentation of this manuscript, the following major revisions should be conducted before publication.

1.       In the abstract , the following two phrases ‘Enzymatically cross-linking of hydrogels possesses many advantages……’ and ‘…overview on cross-linking mechanism of hydrogels…’ are not correct, because hydrogels are already cross-linked.

2.       At line 18, ‘of’ should be ‘or’.

3.       At line 36, RGD should be defined.

4.       At line 67, ‘…in situ gelling hydrogels…’ and at line 87, ‘…In general, in situ gelation of hydrogels…’. These two sentences do not make sense.

5.       At line 134, ‘carboxyl’ should be revised to be ‘carboxylate’. At lines 335 and 524, ‘carboxylic groups’ should be revised to be ‘carboxylic acid groups’. At lines 599 and 633, ‘carboxyl’ should be revised to be ‘carboxy’.

6.       As the section title of 2.1.4, what does ‘thermal condensation7 mean?

7.       In Figure 4, all the sugar structures should be consistently drawn. The structures in Figure 4(b) and (c) are not correct.

8.       In Table 2, enzymatic reactions should be drawn more exactly as those indicate cross-linking process clearly.

9. The conclusion should focus on the subjects on enzymatic cross-linking.

Round 2

Reviewer 1 Report

Dear Authors, I am pleased to inform that your article has been accepted.

Author Response

Thank you for your comments

Reviewer 4 Report

The authors have mostly addressed my comments. For comment 7, however, the sugar structures in Figure 4 are still largely incorrect. Please ask some experts on sugar chemistry to correct them.
